# A Blockchain-Based Hybrid Incentive Model for Crowdsensing

**Lijun Wei**, **Jing Wu** and **Chengnian Long** *

Department of Automation, Shanghai Jiao Tong University, Shanghai 200240, China;
sjtu_weilijun@sjtu.edu.cn (L.W.); jingwu@sjtu.edu.cn (J.W.)
* Correspondence: longcn@sjtu.edu.cn

**Abstract:** Crowdsensing is an emerging paradigm of data aggregation, which has a pivotal role in data-driven applications. By leveraging the recruitment, a crowdsensing system collects a large amount of data from mobile devices at a low cost. The critical issues in the development of crowdsensing are platform security, privacy protection, and incentive. However, the existing centralized, platform-based approaches suffer from the single point of failure which may result in data leakage. Besides, few previous studies have addressed the considerations of both the economic incentive and data quality. In this paper, we propose a decentralized crowdsensing architecture based on blockchain technology which will help improve the attack resistance. Furthermore, we present a hybrid incentive mechanism, which integrates the data quality, reputation, and monetary factors to encourage participants to contribute their sensing data while discouraging malicious behaviors. The effectiveness our of proposed incentive model is verified through a combination of the theory of mechanism design. The performance analysis and simulation results illustrate that the proposed hybrid incentive model is a reliable and efficient mean to promote data security and incentivizing positive conduct on the crowdsensing application.

**Keywords:** crowdsensing; blockchain; incentive mechanism

## 1. Introduction

As an emerging paradigm, crowdsensing plays a vital role in data aggregation [1]. Crowdsensing gathers large amounts of sensing data by utilizing the individual intelligent sensing devices, which effectively reduces the cost and improves the efficiency of data collection [2,3]. With the proliferation of intelligent sensing devices such as cameras, accelerometers, GPSs, etc., and the ongoing expansion of mobile network capabilities, crowdsensing could contribute significantly to the development of the Internet of Things (IoT). This could see crowdsensing applied to transportation [4,5], smart city [6], water-quality monitoring [7], pollution measurement [8,9], human location [10], etc.

Although crowdsensing ideally enhances the efficiency and convenience of data aggregation, there are two significant problems that remain to be solved [11,12]. First, the typical crowdsensing system is mostly managed and maintained by a centralized platform. As shown in Figure 1, the centralized platform which connects the requesters and workers is a key component of the system. However, the centralized system may suffer from the single point of failure [13,14]. Once the central platform is compromised, the sensitive and private data will be at risk of leakage.

Second, in order to encourage workers to contribute their own sensing data, the system must offer appropriate incentive mechanisms to do so. Most of the recent studies focus on the design of a monetary incentive mechanism based on auction theory [15]. It has been demonstrated that the auction-based monetary incentive mechanism is efficient to guarantee that both the requesters and workers benefit. However, the existing monetary incentive mechanisms only provide short-term

incentives, which purely adopts monetary rewards to encourage workers to participate in each time slot. Some passive workers may upload inaccurate data for a selfish purpose in each time slot, resulting in the problem of data quality. Another option is a data-upload-first mechanism [16], which has more of a focus on data quality. This mechanism determines the reward distribution based on the quality of data uploaded by workers. This method improves the quality of sensing data and is more beneficial to requesters. However, this may not really motivate the workers to participate because some workers' benefits will be fewer than the costs.

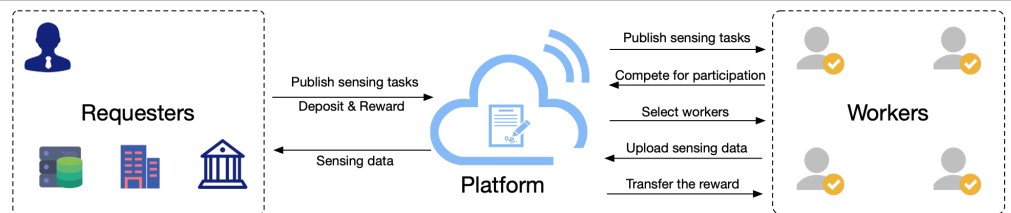

**Figure 1.** The typical crowdsensing architecture.

To address the issues mentioned above, we propose employing the blockchain [17] technology to construct a decentralized crowdsensing architecture. By establishing a consortium based on blockchain, we can effectively prevent the targeted attack. The decentralized architecture afforded by blockchain lacks a single point of failure, making it more fault-tolerant compared to the traditional platform-based methods. Moreover, with the combination of asymmetric encryption and digital signature technology, we will be able to protect individuals' privacy through tight data-access controls [18]. Another major advantage of blockchain-based crowdsensing is that multiparty cooperation means system maintenance costs will be reduced. Specifically, identity authentication and task completion will no longer rely on a central platform. Therefore, unlike in traditional crowdsensing systems, we do not need to construct a dedicated infrastructure to keep the system running.

As this method will lack centralized organization, the process management of sensing task will pose a significant challenge to the decentralized system. To solve this issue, we will employ smart contracts [19] to ensure that each step of a process can be executed automatically, guaranteeing the completion of tasks.

As for the incentive and data quality issues, we propose a hybrid incentive mechanism that comprehensively considers three factors including monetary reward, reputation evaluation, and data quality. After the worker uploads the sensing data, the results of data quality evaluation can be recorded by the requester. This record will be added to the new block. All requesters will be able to calculate and update the workers' reputation by checking the blockchain, which will help requesters to select reliable workers. To amalgamate the three above factors into a convenient evaluation score, we will adopt the Analytic Hierarchy Process (AHP) [20] approach. This will produce a comparable and comprehensive grade which will function as the key criteria for the selection of workers. A further advantage of this model for worker evaluation is that it allows requesters to tailor the calculation of the comprehensive grade to suit their individual needs. Finally, we will apply mechanism design theory to assign a reasonable reward for workers that will maintain the workers' willingness to participate.

There are three key reasons to utilize blockchain for the construction of our incentive model. Primarily, blockchain's decentralization allows for better resistance to system-wide attacks, while simultaneously reducing system operation costs. Secondly, blockchain technology can be used for the long-term preservation of information. This will allow the requester to accurately and regularly assess the credibility of workers by retrieving historic information from the blockchain. Third, based on the behavior information recorded on the blockchain, we propose an incentive mechanism which jointly considers the multiple factors that requesters are most concerned with when they are selecting workers, not only to ensure the incentivization of workers, but also to ensure that requesters can select excellent workers.

The main contributions of this paper are as follows:

- We propose a consortium blockchain-based architecture for crowdsensing, which combines the blockchain and smart contracts to solve the problems of typical crowdsensing systems.
- This is coupled with a hybrid incentive mechanism which integrates monetary reward, reputation, and data quality evaluation. By applying the AHP approach, we can calculate a comprehensive grade for workers introducing greater fairness and practicability to worker selection. Moreover, we adopt the mechanism design theory to design a better reward assignment of workers.
- We simulate the proposed model to verify its feasibility. Our results show that the hybrid incentive mechanism can effectively encourage workers to participate for task completion. Moreover, our method also prevents workers from free-riding and helps requesters to select the most excellent workers.

The remainder of the paper is organized as follows: Related works on the blockchain-based system and incentive mechanism for crowdsensing is presented in Section 2. We then present our system model and detail its operation in Section 3. The hybrid incentive mechanism is presented in Section 4. Simulation results are discussed in Section 5. Section 6 will conclude this paper.

## 2. Related Work

### 2.1. Blockchain Background

Blockchain can be described as a distributed ledger containing a time-stamped series of immutable blocks. It has increasingly found applications in system security protection in recent years [21]. With the combination of blockchain technology, the system can be established without the need of a trusted third-party organization or a central cloud server. Following the success of Bitcoin, blockchain has developed rapidly in recent years [22–25]. Its key attributes are its invulnerability to tampering and its traceability. These characteristics mean it is always possible to determine who is the true owner of something and whether that something is the genuine article. This is what enables blockchain to facilitate the creation of cryptocurrency. Presently, the applications of blockchain technology are constantly expanding, with it currently being introduced into the fields of energy, transportation, medical, etc. [26–32].

Academic research on the blockchain-based crowdsensing system is still in its infancy. [33] proposed a decentralized framework for crowdsourcing based on blockchain. This paper evaluated the time consumption and task cost of applying blockchain in this way. The results supported the effectiveness of blockchain as a tool for crowdsourcing. [34] proposed a crowdsensing quality control model based on blockchain, designing the two-consensus process to achieve a means to evaluate data quality. [35–37] considered the privacy issue by utilizing blockchain. [35] proposed a cooperation verification approach to protect the k-anonymity privacy. [36] constructed the private blockchain network to prevent the identity-based attacks. [37] aimed to handle the location privacy protection by designing the confusion mechanism based on an information coding method.

### 2.2. The Incentive Mechanism of Crowdsensing

The incentive mechanisms in crowdsensing can be mainly divided into two main types: monetary-based mechanism and reputation-based mechanism. The monetary incentive mechanism encourages workers to contribute their sensing data by distributing the rewards. Based on the difference of procedure, [16] divides the monetary incentive into two types: price-decision-first and upload-decision-first. The price-decision-first mechanism, such as [38–41], mostly combines the auction theory [42] to design an optimal mechanism that ensures benefits for requesters and workers. Maximizing social welfare and workers' profit is the primary focus of a price-decision-first mechanism. The upload-decision-first mechanism creates incentives by distributing rewards based on the uploaded data [43–45]. [44] adopted the Expectation Maximization (EM) algorithm to estimate the quality of data for reward assignment. [45] utilized a distance measurement function to estimate data quality

and employed Shapley Value calculations to distribute rewards. There is some use of reputation to select reliable workers in the academic literature [46–50]. Reference [49] employed the fuzzy logic to evaluate the quality of submitted data and constructed a reputation framework for worker selection. Reference [50] adopted the Gompertz function to build the reputation system for crowdsensing.

The above incentive mechanisms are efficient for incentive of crowdsensing. However, most of these studies have suffered from considerable limitations. First, most existing methods rely on a central platform to manage all workers and requesters, which is vulnerable to targeted attack. Second, up to now, little research has been conducted on the multifactor incentive in crowdsensing systems. Existing studies mostly employ single-attribute incentive mechanisms. Although some hybrid incentive mechanisms have been proposed for crowdsensing [51,52], there are still bottleneck problems in usability due to the difficulty of hybrid data management and the need to adjust weightings under a hybrid incentive mechanism. Different from existing incentive mechanisms, our hybrid mechanism is based on consortium blockchain, which has better openness and flexibility for requesters and workers. The bonus of determining the appropriate weighting of factors can be shifted onto the requester, who can adjust the weight of parameters based on their own preferences and needs. By comprehensively considering the behavior of workers and the needs of requesters, our hybrid model helps to achieve a win-win outcome for requesters and workers in the crowdsensing system.

## 3. System Overview

In this section, we outline the model of blockchain-based crowdsensing system and introduce the specific system flow for a sensing task.

### 3.1. A Consortium Blockchain-Based Crowdsensing Framework

A consortium blockchain-based crowdsensing system, as shown in Figure 2, contains two major parties: *requesters* and *workers*. The requester is a task issuer, which acquires sensing data with recruitment. As a data provider, the worker is permitted to perform the sensing task via bidding. The winning worker will be responsible for collecting, uploading the data, and obtaining the corresponding rewards. In blockchain system, messages are defined as the form of "transactions". For example, if the requester $j$ wants to publish a sensing task to the crowd, it will broadcast a transaction like {*Publishing*, *TaskNumber*, *Issuer*, *Content*, *TimeStamp*, *Sig$_j$*}, where *Publishing* is the transaction type. The transaction will invoke the specific smart contract which can be executed automatically. By this way, the requester and the worker can update the state of the sensing task through the combination of smart contract. More importantly, the original transaction will be recorded in the blockchain. It will be nonrepudiation so as to effectively prevent other parties from cheating.

**Requester.** The requester, which may be a research institution, business, or government department, schedules to acquire large amounts of data through the crowd. The consortium is formed by some requesters, and all requesters in the consortium cooperatively maintain the blockchain. As a full node, each requester needs to synchronize the whole blockchain for system operation.

**Worker.** The worker is a data provider in the crowdsensing system. After the authorization of consortium, it can enter the system and receive sensing tasks. The worker is a lightweight node which does not need to synchronize and maintain the blockchain. Moreover, the worker updates the state of task by communicating with the requester.

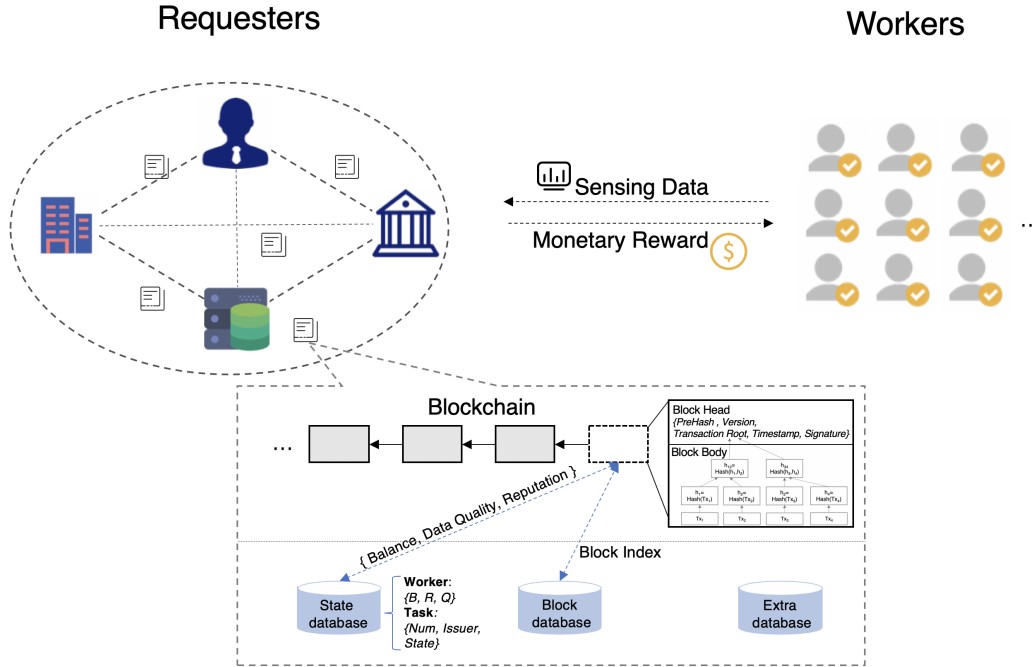

**Figure 2.** The consortium blockchain-based crowdsensing architecture.

*3.2. System Flow*

The overall system flow is shown in Figure 3 and is divided into three parts: *initialization, task process, and system synchronization*. In the part of *initialization*, the necessary configuration and identity authentication mechanisms will be presented. In the part of *task process*, we will introduce the specific steps of crowdsensing. In the part of *system synchronization*, we will discuss and analyze the issue of data and state synchronization.

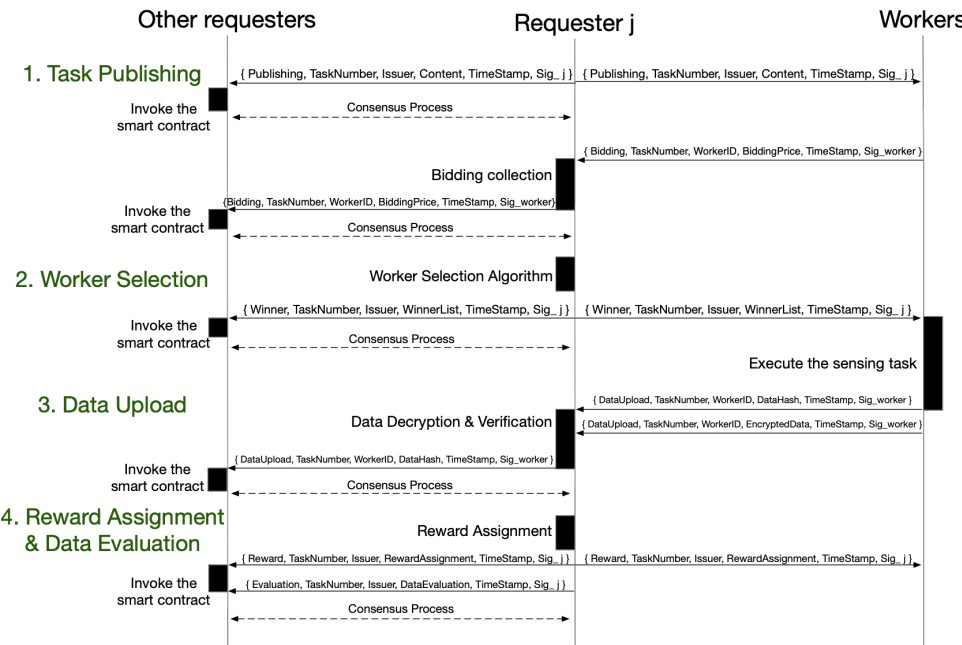

**Figure 3.** The detailed task process in the crowdsensing system.

(1) **System initialization**: In consortium blockchain system, we combine the digital signature scheme and smart contract for system initialization. First, the worker $i$ will send the registration message to any requester for registration. The message content is $\{Register, PII_i, PK_i\}$. The

$PII_i$ denotes the worker's identity information including sensor device brand, property, location, etc. The $PK_i$ denotes the $i$'s public key, which is generated by the worker $i$. After verifying the authenticity of $PII_i$, the requester will generate and send the transaction $RegTx = \{Registration, Hash(PII_i), PK_i, Sig_j\}$ to other requesters. $Hash()$ is the hash function and $Sig_j$ is the signature of requester $j$. Then, $RegTx$ will be packed into the blockchain, which will guarantee the validity of the $RegTx$ transaction. Finally, the requester will return a digital certificate which can be used as the passport of worker $i$ in the crowdsensing system.

(2) **Task process**: The crowdsensing task flow is divided into four steps: task publishing, worker selection, data uploading, and reward assignment and data evaluation.

*Step 1:* A requester $j$ publishes the sensing task by broadcasting the *TaskPublishing* transaction, which is defined as {*Publishing, TaskNumber, Issuer, Content, TimeStamp, $Sig_j$*}. The transaction will invoke the smart contract to update the task state. The *TaskPublishing* transaction will be appended into blockchain after the consensus between requesters.

*Step 2:* After receiving the *TaskPublishing* message, workers will decide whether to compete for participation and submit the bidding price by sending the *Bidding* transaction, which is defined as {*Bidding, TaskNumber, WorkerID, BiddingPrice, TimeStamp, $Sig_{worker}$*}. The *BiddingPrice* reflects their costs of performing the sensing task. Then, the requester will consider the worker's reliability comprehensively based on workers' bidding price, reputation, and recent data quality to select the appropriate workers. The specific worker selection method will be introduced in Section IV.

*Step 3:* After receiving the result of worker selection, the winning workers will perform the task and upload the sensing data. In order to protect the data privacy, we will combine the AES (Advanced Encryption Standard) and public key encryption to achieve the authorization of data access. Specifically, the worker first encrypts the data with a symmetric key, then encrypts the symmetric key with the public key of the specific requester. After the requester receives the message, it first decrypts the message with its own private key and gets the symmetric key. Then, it will decrypt the data file with the symmetric key. In order to reduce the storage pressure of data in the blockchain, we only save the hash of the uploaded data in the blockchain.

*Step 4:* In this step, the requester will distribute the workers' rewards and evaluate the data quality. The data evaluation result will be recorded in the blockchain and influence the worker's reputation value. The detailed message or transaction content is shown in Figure 3.

(3) **State synchronization**: Due to the absence of a central platform for managing task process, we utilize blockchain to synchronize the state update about tasks and workers. The requester will pack all valid transactions which include the registration and task information into the new block. Then, it will send the new block to other requesters. After performing the specific consensus protocol, the new block will be added to the blockchain, which guarantees the long-term validity of transactions. Any requester can acquire historical block data by synchronizing the blockchain. There are already many effective and secure consensus protocols such as PoW [17], PBFT [53], Raft [54], PoS [55], etc. A complete description of consensus protocol is beyond the scope of this paper.

## 4. Hybrid Incentive Mechanism Design

Two major problems will be encountered in the worker selection and reward assignment steps. The requester has no knowledge about workers' data quality before the data has been uploaded. Therefore, workers can easily upload misleading sensing data. Additionally, an unreasonable reward distribution may reduce workers' enthusiasm. In this section, we present the hybrid incentive mechanism to address the two issues. The factors of reputation and recent data quality are considered for first problem. The reward assignment method, which is based on bidding price and comprehensive grade,

is used to achieve fair reward distribution. For the sake of reading, Table 1 lists the notations and descriptions frequently used in the paper.

**Table 1.** Notations and descriptions

| Notations | Descriptions |
|---|---|
| $W$ | The set of all workers |
| $J$ | The set of all requesters |
| $Z$ | The set of winning workers |
| $z$ | The number of winning workers |
| $n$ | The number of workers |
| $\theta_i$ | The comprehensive grade of worker $i$ |
| $\theta_{-i}$ | The set of comprehensive grades of workers other than worker $i$ |
| $b_i$ | The bidding price of worker $i$ |
| $c_i$ | The cost price of worker $i$ |
| $B_i$ | The bidding rank of worker $i$ |
| $R_i$ | The reputation of worker $i$ |
| $Q_i$ | The recent data quality of worker $i$ |
| $(\omega_1, \omega_2, \omega_3)$ | The weighted value of bidding price, reputation, and recent data quality |
| $m$ | The number of transactions in each block |
| $h$ | The block height[1] |
| $h_0$ | The current block height |
| $H_1$ | The set of blocks which is considered for reputation calculation, where the elements in $H_1$ satisfy $h_0 - h \leq |H_1|$ |
| $H_2$ | The set of blocks which is considered to recent data quality calculation, where the elements in $H_2$ satisfy $h_0 - h \leq |H_2|$ |
| $\lambda_1$ | The aging parameter of satisfactory evaluation of workers' sensing data |
| $\lambda_2$ | The aging parameter of negative evaluation of workers' sensing data |
| $q_{i,h}(k)$ | The data quality rating of the $i$'s task $k$ at the block height $h$ |
| $K_i$ | The set of tasks which are performed by worker $i$ |

[1] Block height is defined as the number of blocks preceding a particular block in the blockchain.

The hybrid incentive mechanism is based on three important parameters including workers' bidding, reputation, and recent data quality estimation. Despite the fact that the hybrid incentive can overcome the drawbacks of traditional incentives, this idea brings up a new problem; that is, how to integrate the above three independent variables. To solve the multifactor decision problem, we employ the AHP method. $\theta$ is calculated by the weighted sum of workers' bidding, reputation, and recent data quality. The worker with larger $\theta$ has a higher probability of being selected. Thus, we have the following $\theta$ calculation formula:

$$\theta_i = \omega_1 B_i + \omega_2 R_i + \omega_3 Q_i, \tag{1}$$

where $\omega_i \geq 0$ and $\sum_{\omega_i=1}^{3} \omega_i = 1$. Setting the weighted value directly is difficult for a requester, and an inexperienced requester may not be able to select optimal workers due to an incorrect weight setting. Therefore, in order to balance the sensing cost and data quality, we use the AHP method to calculate $\omega_1$, $\omega_2$, and $\omega_3$. The specific framework is as shown in Figure 4.

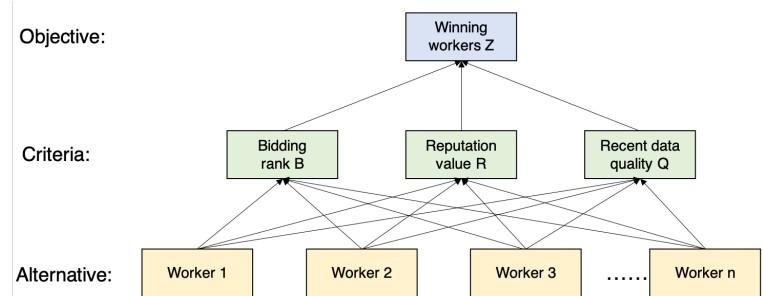

**Figure 4.** The hierarchical structure of worker selection.

The framework is composed of three levels, the alternative leveL, the criteria level, and the objective level. In order to choose the appropriate workers, we take the winning worker set as the objective; the three factors including bidding rank, reputation, and recent data quality as the criteria level; and $n$ workers as candidates. Next, we describe in detail how to calculate the three factors.

*4.1. The Calculation of Three Factors*

**Bidding price**: When receiving the *TaskPublishing* message, workers will estimate the cost of performing the sensing task and submit the bidding price $b_i$. A lower bidding price is more satisfying to requesters. The worker with a lower bidding price has a higher probability of being selected. In order to unify the scale of three factors, we scale the $B_i = 1 - \frac{b_i - \underline{b}}{\overline{b} - \underline{b}}$, where $\overline{b}$ and $\underline{b}$ are the maximum and minimum of bidding price. Thus, we have $\theta_i = \gamma(b_i, R_i, Q_i) = \omega_1(1 - \frac{b_i - \underline{b}}{\overline{b} - \underline{b}}) + \omega_2 R_i + \omega_3 Q_i$, where only $b_i$ is decided by worker $i$ in the worker selection step.

**Reputation value**: We adopt a reputation engine to calculate the worker's reputation. When the requester receives the sensing data, it evaluates the data quality and records the feedback in the message which will ultimately be appended to the new block (as shown in Figure 3). Any requester can access the information of data quality evaluation about each worker. Since our goal is to build a general crowdsensing system, different requesters may have different data requirements such as accuracy, geographic location, etc. It is so difficult to design a unified data quality evaluation algorithm which is adaptive to all crowdsensing scenarios. Therefore, we use the rank evaluation method to rate the data quality. The rating is divided into three levels: satisfactory, medium, and negative. We make $q_{i,h}(k)$ equal to 1 if the quality rating is satisfactory. Similarly, we make $q_{i,h}(k)$ equal to 0 if the quality rating is medium and -1 if the quality rating is negative. Based on data quality rating, we design a specific reputation engine by employing the classical beta-binomial framework to calculate the reputation value of workers. Reputation engine based on beta function is feasible due to its flexibility and simplicity, which will save more computation cost of reputation calculation. The beta distribution is expressed using the gamma function as

$$p(x) = \frac{\Gamma(\alpha + \beta)}{\Gamma(\alpha)\Gamma(\beta)} x^{\alpha - 1}(1 - x)^{\beta - 1}, \tag{2}$$

where $0 \le x \le 1, \alpha \ge 0, \beta \ge 0$. By combining quality rating, we will apply the beta function to our reputation engine. The medium evaluation will not influence the worker's reputation. We define $m_{i,h}^j$ as the amount of satisfactory evaluation of worker $i$'s sensing data, denoted by requester $j$ at the block height $h$. Similarly, we define $g_{i,h}^j$ as the amount of negative evaluation. Then, the reputation function in our model can be written as

$$p(x) = \frac{\Gamma(M_i + G_i + 2)}{\Gamma(M_i + 1)\Gamma(G_i + 1)} x^{M_i}(1 - x)^{G_i}, \tag{3}$$

where $M_i = \sum_{j \in J, h \in H_1} m_{i,h}^j \lambda_1^{h_0 - h}$ and $G_i = \sum_{j \in J, h \in H_1} g_{i,h}^j \lambda_2^{h_0 - h}$. By using Equation (3), the current reputation of worker $i$ can be calculated based on probability expectation value of the reputation function:

$$R_i = \frac{M_i + 1}{M_i + G_i + 2}. \tag{4}$$

**Recent data quality**: We use the "recent data quality" to evaluate the quality of recently uploaded data, which reflects the latest behaviors of workers. It is a complementary factor of long-term effect of reputation. Without this factor, a short-term evil by a worker with a high reputation value will not substantially affect its reputation value and comprehensive grade. Therefore, in order to avoid such a situation, we have incorporated the factor of recent data quality into the comprehensive assessment of workers. The specific calculation is as follows:

$$Q_i = \frac{\sum_{h \in H_2, k \in K_i} q_{i,h}(k) - \underline{Q}}{\overline{Q} - \underline{Q}}, \tag{5}$$

where $\overline{Q}$ and $\underline{Q}$ are the maximum and minimum of $Q$.

### 4.2. A Multifactor Worker Evaluation Approach

The worker $i$'s grade $\theta_i$ can be calculated based on $B_i$, $R_i$, and $Q_i$. We use the AHP to calculate the weighted value $\omega_1$, $\omega_2$, and $\omega_3$. By combining the 1–9 scale method proposed by Saaty [20], we obtain the pairwise comparison matrix $A$ as follows:

$$A = \begin{bmatrix} a_{11} & a_{12} & a_{13} \\ a_{21} & a_{22} & a_{23} \\ a_{31} & a_{32} & a_{33} \end{bmatrix} = \begin{bmatrix} B/B & B/R & B/Q \\ R/B & R/R & R/Q \\ Q/B & Q/R & Q/Q \end{bmatrix}. \tag{6}$$

We then calculate the normalized pairwise comparison matrix $\overline{A}$ as follows:

$$\overline{A} = \begin{bmatrix} \overline{a}_{11} & \overline{a}_{12} & \overline{a}_{13} \\ \overline{a}_{21} & \overline{a}_{22} & \overline{a}_{23} \\ \overline{a}_{31} & \overline{a}_{32} & \overline{a}_{33} \end{bmatrix} = \begin{bmatrix} \frac{a_{11}}{\sum_{i=1}^3 a_{i1}} & \frac{a_{12}}{\sum_{i=1}^3 a_{i2}} & \frac{a_{13}}{\sum_{i=1}^3 a_{i3}} \\ \frac{a_{12}}{\sum_{i=1}^3 a_{i1}} & \frac{a_{22}}{\sum_{i=1}^3 a_{i2}} & \frac{a_{23}}{\sum_{i=1}^3 a_{i3}} \\ \frac{a_{13}}{\sum_{i=1}^3 a_{i1}} & \frac{a_{32}}{\sum_{i=1}^3 a_{i2}} & \frac{a_{33}}{\sum_{i=1}^3 a_{i3}} \end{bmatrix}. \tag{7}$$

Based on the normalized pairwise comparison matrix, the weighted value can be obtained by calculating the arithmetic mean of each row. That is,

$$(\omega_1, \omega_2, \omega_3) = \frac{1}{3}\left(\sum_{j=1}^3 \overline{a}_{1j}, \sum_{j=1}^3 \overline{a}_{2j}, \sum_{j=1}^3 \overline{a}_{3j}\right). \tag{8}$$

Based on AHP, the requester can obtain the weighted value by setting the comparison matrix. The weighted value is different and depends on the requester's requirements. For instance, if the requester $j$ deems the reputation is more important than the other two factors, then it will set a larger $R/B$ and $R/Q$. Thus, the weighted value of reputation $\omega_2$ will be larger. In addition, we need to emphasize that when we use the 1–9 scale method to design comparison matrix $A$, the consistency test of $A$ is necessary to ensure the effectiveness of AHP.

### 4.3. Worker Selection and Reward Assignment

For each task, we assume the requester needs to recruit $z$ workers. The first key problem here is how to select the appropriate workers. Furthermore, in order to motivate workers to contribute their sensing data, we must assign the suitable reward to winning workers. Therefore, the second key problem is how to distribute the rewards to the workers. In this section, we combine the mechanism design theory [56] to guarantee two important properties of our incentive mechanism: incentive

compatibility and individual rationality, thereby improving the effectiveness of worker selection and reward assignment.

The two main desired properties of designing worker selection and reward assignment mechanisms are as follows.

1. **Incentive compatibility (IC)**: The truthful submission of sensing cost is the worker's optimal bidding strategy. In other words, each worker will submit the sensing cost as its bidding price.
2. **Individual rationality (IR)**: The reward must compensate for the worker's cost, that is, the worker's utility should be non-negative when the worker truthfully submits the bidding price.

We note that the worker $i$ is uncertain about other workers' grades in our crowdsensing system. Therefore, the problem is an incomplete information game for workers. We shall assume that each $\theta_i$ is independently and identically distributed on interval $[\underline{\theta}, \overline{\theta}]$ according to the uniform distribution function $F$. We let $f$ be the probability density function of $F$. Then, $f(\theta_i) = \frac{1}{\overline{\theta} - \underline{\theta}}$, where $\underline{\theta} \leq \theta_i \leq \overline{\theta}$.

We denote by $p_i(\theta_i)$ the probability function that the worker $i$ is selected by the requester. We denote by $t_i(\theta_i)$ the reward function of the worker $i$. We denote by $c_i$ the cost of the worker $i$. If the worker $i$ declares its bidding price which is equal to the cost, $b_i = \gamma^{-1}(\theta_i) = c_i$, this is equivalent to the true $\theta_i$ is submitted. Meanwhile, other workers submit their true $\theta_{-i}$ by truthfully declaring their bidding price $b_{-i}$. Thus, the conditional expected probability that worker $i$ will be selected can be defined as

$$P_i(\theta_i) = \eta(p_i(\theta_i, \theta_{-i})) = \int_{\theta_{-i}} p_i(\theta_i, \theta_{-i}) f_{-i}(\theta_{-i}) d\theta_{-i}. \tag{9}$$

Similarly, the conditional expected value of the reward that the worker $i$ will obtain can be given by

$$T_i(\theta_i) = \zeta(t_i(\theta_i, \theta_{-i}) = \int_{\theta_{-i}} t_i(\theta_i, \theta_{-i}) f_{-i}(\theta_{-i}) d\theta_{-i}. \tag{10}$$

Therefore, the conditional expected value worker $i$'s utility can be calculated as

$$U_i(\theta_i) = T_i(\theta_i) - P_i(\theta_i)c_i. \tag{11}$$

Each worker $i$ strategically submits the $\theta_i$ by declaring $b_i$ to maximize its $U_i$. According to the definitions of IC and IR, the condition of IC in our model can be expressed as follows:

$$U_i(\theta_i|\theta_i) \geq U_i(\theta_i'|\theta_i), \tag{12}$$

and, for IR, it is

$$U_i(\theta_i) \geq 0, \forall \theta_i \in [\underline{\theta}, \overline{\theta}]. \tag{13}$$

We combine the Myerson's well-known theorem [56,57] to design the complete worker selection and reward assignment method:

**Worker selection**:

$$p_i(\theta_i, \theta_{-i}) = \begin{cases} 1, & if \ \theta_i > \theta_{(n-z)} \\ \\ 0, & otherwise, \end{cases} \tag{14}$$

**Reward assignment**:

$$t_i(\theta_i, \theta_{-i}) = \chi(t^*) = \begin{cases} t^*, & if \ p_i(\theta_i) = 1 \\ \\ 0, & otherwise, \end{cases} \tag{15}$$

where $t^* = \gamma^{-1}(\theta_i) + \frac{\int_{\underline{\theta}}^{\theta_i} \frac{\overline{b} - \underline{b}}{\omega_1} P_i(x) dx}{P_i(\theta_i)}$ is the reward value to worker $i$ when the worker is selected. $\theta_{(n-z)}$ is referred to as the $(n-z)$-highest-order statistic. Therefore, the set of winning workers $Z$ can be written as $Z = \{i|\theta_i > \theta_{(n-z)}\}$.

**Theorem 1.** *The mechanism (p, t) is incentive-compatible and individual-rational.*

**Proof.** We first prove the condition of IC. According to (10) and (15), we have

$$T_i(\theta_i) = \zeta(t_i(\theta_i, \theta_{-i}) = P_i(\theta_i)\gamma^{-1}(\theta_i) + \int_{\underline{\theta}}^{\theta_i} \frac{\overline{b} - \underline{b}}{\omega_1} P_i(x)dx. \tag{16}$$

Considering that two types $\theta_i$ and $\theta_i'$, where $\theta_i \neq \theta_i'$, we have

$$
\begin{aligned}
&[T_i(\theta_i) - P_i(\theta_i)c_i] - [T_i(\theta_i') - P_i(\theta_i')c_i]\\
&= [\int_{\underline{\theta}}^{\theta_i} \frac{\overline{b}-\underline{b}}{\omega_1} P_i(x)dx] - [P_i(\theta_i')\gamma^{-1}(\theta_i') + \int_{\underline{\theta}}^{\theta_i'} \frac{\overline{b}-\underline{b}}{\omega_1} P_i(x)dx + P_i(\theta_i')c_i]\\
&= \int_{\theta_i'}^{\theta_i} \frac{\overline{b}-\underline{b}}{\omega_1} P_i(x)dx - P_i(\theta_i')(\gamma^{-1}(\theta_i') - \gamma^{-1}(\theta_i))\\
&= \frac{\overline{b}-\underline{b}}{\omega_1}[\int_{\theta_i'}^{\theta_i} P_i(x)dx - P_i(\theta_i')(\theta_i - \theta_i')]\\
&\geq 0.
\end{aligned}
\tag{17}
$$

The last inequality is valid due to the nondecrease of $P_i(\theta_i)$. Therefore, according to the definition in (12), mechanisms (14) and (15) satisfy the condition of IC.

We next prove the condition of IR. According to (11), (14), and (15), we have

$$U_i(\theta_i) = P_i(\theta_i)\gamma^{-1}(\theta_i) + \int_{\underline{\theta}}^{\theta_i} \frac{\overline{b}-\underline{b}}{\omega_1} P_i(x)dx - P_i(\theta_i)c_i = \int_{\underline{\theta}}^{\theta_i} \frac{\overline{b}-\underline{b}}{\omega_1} P_i(x)dx. \tag{18}$$

Then, we have

$$U_i(\underline{\theta}) = 0 \quad and \quad U_i'(\theta_i) = \frac{\overline{b}-\underline{b}}{\omega_1} P_i(\theta_i) \geq 0. \tag{19}$$

Therefore, $U_i(\theta_i) \geq 0, \forall \theta_i \in [\underline{\theta}, \overline{\theta}]$. Mechanisms (14) and (15) satisfy the condition of IR. □

## 5. Simulation and Results

In order to validate the effectiveness of our proposed model, we have performed a number of vital tests to measure the performance of our model. We use *round* to represent a time unit. Only one block will be generated in each time unit. Besides, the number of transactions in each block is settled in the range of 10 $\sim$ 60. The specific parameters we use in the experiment are listed in Table 2. Among these, the weighted value $(\omega_1, \omega_2, \omega_3)$ is derived by the design of comparison matrix $A$. This section is divided into three parts. First, we divide the workers into different groups and compare the overall performance of different groups. Second, we study the change of reputation, balance, and profit of different workers. Finally, we analyze the storage overhead of blockchain data.

**Table 2.** Parameters and Settings.

| Parameter | Setting |
|---|---|
| $n$ | 100 |
| $A$ | $\begin{bmatrix} 1 & 6 & 8 \\ 1/6 & 1 & 3 \\ 1/8 & 1/3 & 1 \end{bmatrix}$ |
| $(\omega_1, \omega_2, \omega_3)$ | $(0.753, 0.172, 0.075)$ |
| $\lambda_1$ | 0.96 |
| $\lambda_2$ | 0.995 |
| $z$ | $(10,12)$ |
| $m$ | $10 \sim 60$ |
| $|H_1|$ | 100 |
| $|H_2|$ | 20 |

## 5.1. Overall Evaluation of Different Groups

We mainly test the results of workers' reputation value, balance, and profit to verify the effectiveness of our incentive model. In order to compare the credibility and benefit of workers with different behavioral characteristics, we divide all workers into four groups:

**Group 1**: Workers "No.1 $\sim$ 25" are excellent workers that always upload valid sensing data.

**Group 2**: Workers "No.26 $\sim$ 50" are negligent workers that may upload false data because of accidental reasons such as device crash with a probability of 10% $\sim$ 20%.

**Group 3**: Workers "No.51 $\sim$ 75" are indifferent workers that do not care about their reputation. They upload false sensing data with a probability of 40% $\sim$ 60%.

**Group 4**: Workers "No.76 $\sim$ 100" are malicious workers that always upload false data for saving the sensing cost and getting more profit.

We compare the average reputation, balance, and profit of different groups. As shown in Figure 5a and Figure 6a, we note that the workers' reputation will be enhanced as they become more truthful and reliable. The reputation of the excellent workers will gradually increase. They will be more likely to be selected as winning workers and obtain more profits, as shown in Figure 5c and Figure 6c. On the other hand, the reputation of malicious workers will gradually decrease and the workers will be isolated and obtain zero profit ultimately. Figure 5b shows that the negligent and indifferent workers have chances to perform sensing tasks at the early stage. However, since they sometimes upload false data, after a while, the requesters will prefer excellent workers, as shown in Figure 6b. This demonstrates that more excellent workers will be selected as the winning workers to perform the sensing task, which will help improve the overall data quality obtained by the requester.

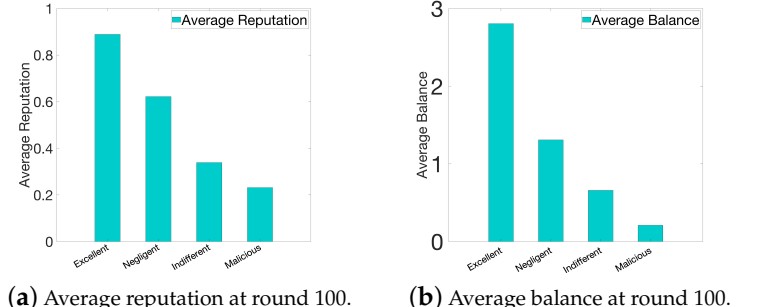

(**a**) Average reputation at round 100.　　(**b**) Average balance at round 100.　　(**c**) Average profit at round 100.

**Figure 5.** The reputation, balance, and profit of different groups at round 100.

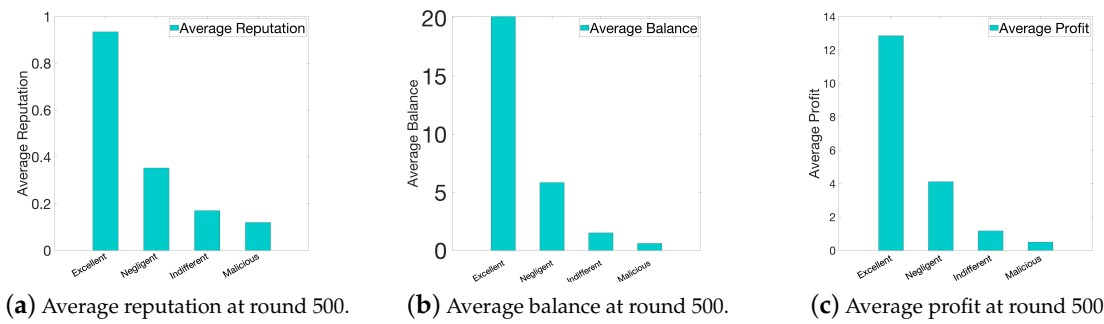

(**a**) Average reputation at round 500.　　(**b**) Average balance at round 500.　　(**c**) Average profit at round 500.

**Figure 6.** The reputation, balance, and profit of different groups at round 500.

## 5.2. Detailed Comparison of Different Workers

After an overall comparison, we further choose one worker from each group and study their changes of reputation, balance, and profit. Specifically, we choose the excellent worker with high sensing costs, and the negligent worker and indifferent worker with low sensing costs.

As shown in Figure 7a, $Quality = 1$ represents the case when the requester gives the worker a satisfactory evaluation and vice versa. $Quality = 0$ indicates that the quality of the uploaded data is medium or the worker is not selected.

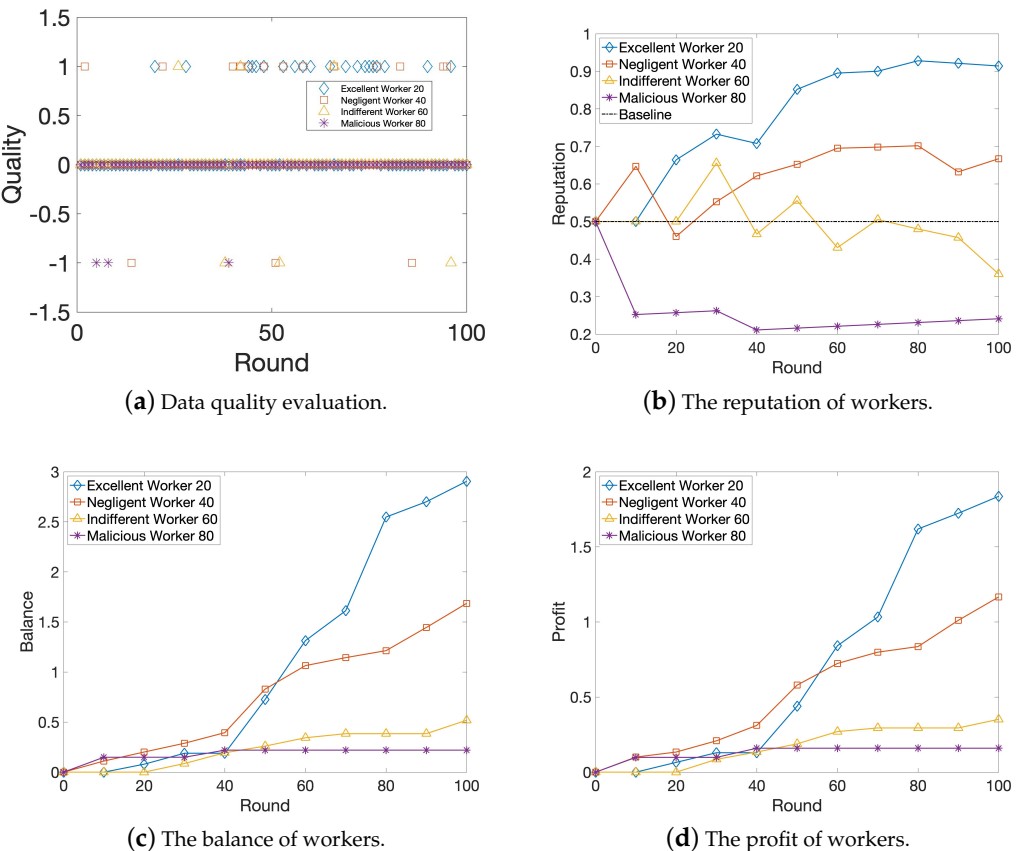

(**a**) Data quality evaluation.　　　　　　　　　(**b**) The reputation of workers.

(**c**) The balance of workers.　　　　　　　　　(**d**) The profit of workers.

**Figure 7.** The reputation, balance, and profit of different types of workers.

Figure 7b shows that the excellent worker will receive better evaluation when it uploads high-quality data continuously. Thus, its reputation value will continue to increase. Due to the high cost of performing the task, the likelihood of excellent worker 20 being selected by the requester is low at the early stage. However, along with the rise of its reputation and high quality of data it

provides, the proportion of worker 20 being selected will increase gradually. Moreover, if the worker tries to upload false data for cost saving, its reputation value will gradually decrease. Therefore, this demonstrates that the truthful and untruthful workers can be identified based on our reputation engine.

Figure 7c,d shows the balance and profit of different workers. It is noted that the excellent worker 20 performs more sensing tasks than the other three types of workers and obtains more rewards as the growth of $R_{20}$ and $Q_{20}$. The profit of worker 20 will gradually become more than others. This shows that our incentive model can effectively motivate excellent workers to participant and contribute the sensing data. Meanwhile, it prevents malicious workers from uploading false data to disrupt the crowdsensing system.

Next, we verify whether our incentive model can prompt workers to truthfully submit their own cost as the bidding price. We let the worker 10 always truthfully submit its bidding price. Additionally, we let worker 11's sensing cost $c_{11}$, reputation $R_{11}$, and recent data quality $Q_{11}$ be equal to that of worker 10. However, worker 11 is a selfish worker that strategically submits bidding price which deviates the price of its own cost. As shown in Figure 8, worker 11 submits the bidding at a price 1% higher than the cost. As the bidding price increases, the probability of worker 11 being selected decreases. Although its profit will increase once it is selected, the total profit of worker 11 will still be lower than worker 10. The results of the experiment also confirm this. Figure 9 shows that worker 11 submits the bidding at a price 1% lower than the cost. Although worker 11 will be more likely to be selected, it gets less profit than worker 10. This demonstrates that the optimal bidding strategy of the worker is submitting the bidding price which is equal to the cost.

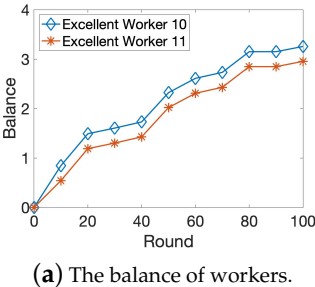

(**a**) The balance of workers.

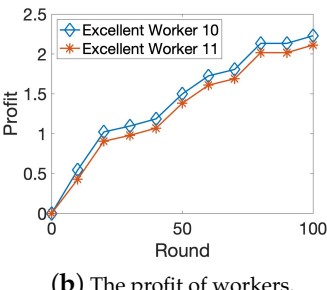

(**b**) The profit of workers.

**Figure 8.** The comparison of worker 10 and worker 11 that submits the bidding at a price 1% higher than the cost.

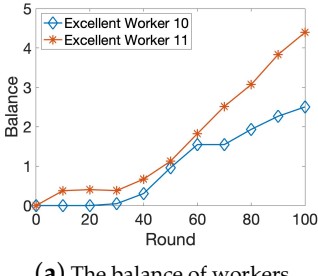

(**a**) The balance of workers.

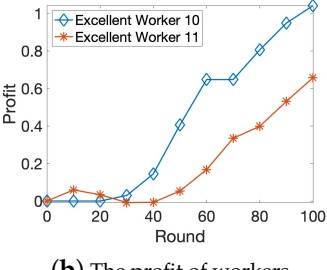

(**b**) The profit of workers.

**Figure 9.** The comparison of worker 10 and worker 11 that submits the bidding at a price 1% lower than the cost.

### 5.3. Storage Overhead

We will discuss the storage overhead of blockchain data in this part. A block header is about 80 bytes [17]. The data in the block consists of two parts: *RegTx* (about 150 bytes) and the task related transaction (about 150 ∼ 200 bytes) such as *TaskPublishing*, *Bidding*, etc. The frequency

of block generation will be settled within 1 minute. The storage overhead of block head and task related transaction in one block is $(80 + (150 \sim 200) * m)$ bytes, where $m$ is the number of transactions in the block. Therefore, the storage overhead of the whole blockchain is about $150 \cdot z + (80 + (150 \sim 200) * m) * h_0$ bytes. The detailed storage overhead of blockchain is shown in Figure 10. The result has demonstrated that the storage cost of synchronizing blockchain is affordable to requesters. The effective method for lightening the storage burden will be considered as our future work.

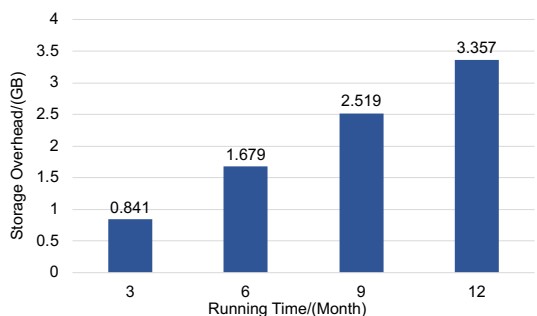

**Figure 10.** The storage overhead of blockchain.

## 6. Conclusions

In this paper, we propose a consortium blockchain-based incentive model for the crowdsensing system. On the one hand, by combining the consortium blockchain technology, we can construct a crowdsensing system that is resistant to the single point of failure. Cooperative management will reduce the cost and enhance the flexibility of the system. Moreover, the use of blockchain means the security of the crowdsensing system will not need to rely on a trusted third-party organization. The system will be more open and flexible, enhancing the feasibility and practicality of our proposed model for numerous crowdsensing applications such as indoor localization, transportation, health monitoring, etc. On the other hand, the hybrid incentive mechanism is proposed to encourage workers to contribute valuable data while penalizing the malicious workers. Compared to the traditional incentive mechanisms, which only consider the single attribute when determining appropriate rewards, our hybrid mechanism considers the monetary incentives, data quality control, and reputation simultaneously to ensure fairness and prevent fraud. Experimental results illustrate the efficiency of our proposed model and demonstrate that the hybrid incentive model ensures favorable short-term and long-term incentives for workers.

Further research investigating the efficacy and safety of blockchain-based incentive models would be significant. Additionally, there are a number of questions posed by this research that warrant further investigation. First, the proposed hybrid incentive model only considers a static situation where the evaluation attributes for workers will not be changed. In the future, we will consider more attributes which are related to workers' credibility and focus on establishing a more adaptive hybrid incentive model to overcome new problems caused by dynamic changes in the demand for crowdsensing tasks. Second, the optimization of consensus protocol of blockchain is our interest direction. The consensus protocol greatly affects the performance of the blockchain-based system. Although there are many practical consensus protocols such as PBFT, Raft, etc., crowdsensing-oriented optimization has yet to be considered and should be studied further. Third, a means to further protect worker privacy, particularly when calculating their comprehensive grade, will need to be investigated.

**Author Contributions:** Conceptualization, L.W.; methodology, L.W. and J.W.; software, L.W.; validation, L.W., J.W., and C.L.; formal analysis, L.W.; investigation, L.W.; writing–original draft preparation, L.W. and J.W.; writing–review and editing, L.W., J.W., and C.L.; supervision, C.L.; project administration, C.L.; funding acquisition, C.L. All authors have read and agreed to the published version of the manuscript.

**Funding:** This work was supported in part by the National Nature Science Foundation of China under Grants 61673275, 61873166 and Science and Technology Commission of Shanghai Municipality under Grant 19511102102.

**Conflicts of Interest:** The authors declare no conflict of interest.

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
