# Peer review of "A Blockchain-Based Hybrid Incentive Model for Crowdsensing"

_electronics, doi:10.3390/electronics9020215_

Round 1

Reviewer 1 Report

The paper clearly presents issues related to the mass collection of data from mobile devices, highlighting critical issues such as security, privacy protection and encouraging data collection activity. The authors present a decentralized blockchain-based technology for mass
sensing data collection in order to avoid security risk.
The paper lacks an overview of existing hybrid approaches to address the
problem under consideration. The authors are asked to present existing approaches.

Suggested corrections:

Line 21: please delete „Despite that“ at the beginning of the sentence, because it is confusing.

Line 94: please correct „crowdsourcing“

Line 180: please move figure 3. after the point at which you mention the figure in the text. It is confusing when you get to the figure without referencing it first.

Reviewer 2 Report

Dear Authors.

This paper (A Blockchain-based Hybrid Incentive Model for Crowdsensing) provides little information. I don't recommend publication right now. But, it is good motivation and result. So, I give Major Revision.

#1.
I recommend additional/rewrite "Abstract and contribution".

#2.
Introduction provide sufficient background and include all relevant references

#3.
What is Blockchain-based Hybrid Incentive Model?

What is Blockchain?

Also, improve the related works (Deep Neural Network, Microgrid), an important aspect has been mentioned:
e.g. [1], [2], [3], [4], [5], [6], [7].

[1] "The truth about blockchain." Harvard Business Review 95.1 (2017): 118-127.

[2] "A study on improvement of blockchain application to overcome vulnerability of IoT multiplatform security." Energies 12.3 (2019): 402.

[3] "Bitcoin-ng: A scalable blockchain protocol." 13th USENIX Symposium on Networked Systems Design and Implementation, 2016.

[4] "Blockchain challenges and opportunities: A survey." International Journal of Web and Grid Services 14.4 (2018): 352-375.

[5] "An overview of blockchain technology: Architecture, consensus, and future trends." 2017 IEEE International Congress on Big Data (BigData Congress). IEEE, 2017.

[6] "Blockchain-based mobile fingerprint verification and automatic log-in platform for future computing." The Journal of Supercomputing 75.6 (2019): 3123-3139.

[7] "Where is current research on blockchain technology?—a systematic review." PloS one 11.10 (2016): e0163477.

#4.
I recommend additional/rewrite "Introduction".

#5.
If you have time "Conclusion and future work" write more.

#6.
There are several issues in the layout and references.

#7.
Scientific papers should be replicable.

Reviewer 3 Report

The paper proposes a consortium blockchain-based architecture for mobile crowdsensing (MCS) in order to exploit smart contracts and blockchains to tackle typical MCS issues. The proposal is enriched with a hybrid incentive mechanism for MCS participants. The model's feasibility has been validated via simulation.

The manuscript is well-written and methodologically sound. Its structure is advisable and presents an interesting perspective about how to tackle sigle point of failures in centralised MCS systems.

Suggestions:

1) improving literature review on MCS systems, by specifying current solutions deal with several different sectors (see references reported below) as only acoustic pollution monitoring and road condition control have been referenced.

2) evaluating whether the proposed approach is more suitable to some MCS application domains than other ones

3) reduce the size of Fig.10, since it is disproportionate if compared to other figures.

For these reasons, a suggest a minor review of the manuscript before accepting it for publication.

Suggested additional bibliographical references:

[1] Longo, A., De Matteis, A., Zappatore, M., Urban pollution monitoring based on Mobile Crowd Sensing: an osmotic computing approach. Proc. of 2018 IEEE 4th Int. Conf. on Collaboration and Internet Computing (CIC), pp. 380-387, 2018.

[2] Minkman, E., Van Overloop, P.J., Van der Sanden, M., Citizen Science in Water Quality Monitoring: Mobile Crowd Sensing for Water Management in the Netherlands. Proc. of World Environmental and Water Resources Congress 2015, pp. 1399-1408, 2015.

[3] Liu, K., Li, X., Finding nemo: Finding your lost child in crowds via mobile crowd sensing. Proc. of 2014 IEEE 11th Int. Conf. on Mobile Ad Hoc and Sensor Systems, pp. 1-9, 2014.

[4] Longo, A., Zappatore, M., Bochicchio, M.A., Towards massive open online laboratories: an experience about electromagnetic crowdsensing. Proc. of 2015 12th Int. Conf. on Remote Engineering and Virtual Instrumentation (REV2015), pp. 43-51, 2015.

[5] Gil, D.S., D'Orey, P.M., Aguiar, A., On the challenges of mobile crowdsensing for traffic estimation. Proc. of the 15th ACM Conf. on Embedded Network Sensor Systems SenSys '17, pp.52:1-52:2, 2017.

[6] Chowdhury, C., Roy, S., Mobile Crowd-Sensing for Smart Cities, Wiley-Blackwell, pp. 125-154, 2017.

Round 2

Reviewer 2 Report

Dear Authors.

The revision adequately address the concerns expressed in last review.
So, I recommend that this revised manuscript can now be recommended for publication (Accept as is: Accept in present form).